# Hygrothermal Effect on GF/VE and GF/UP Composites: Durability Performance and Laboratory Assessment

**DOI:** 10.3390/polym16050632

**Published:** 2024-02-26

**Authors:** Dengxia Wang, Yan Sun, Jian Duan, Keyong Xie, Jikai Li, Qi An, Xinbo Wang

**Affiliations:** 1The 7th Research Department, Shandong Institute of Non-Metallic Materials, Tianzhuang East Road No. 3, Jinan 250031, China13011092066@163.com (J.L.); 17865164001@163.com (Q.A.);; 2Institute of Laser & Optoelectronics, School of Precision Instruments and Opto-Electronics Engineering, Tianjin University, Tianjin 300072, China

**Keywords:** glass fiber-reinforced composites, degradation mechanism, failure model, unsaturated polyester, vinyl ester

## Abstract

In order to investigate the durability of two kinds of fiber-reinforced composite materials, and obtain the degradation mechanism and failure model in a hygrothermal environment, E-glass- fiber-reinforced composite materials, glass fiber-reinforced epoxy vinyl ester and glass fiber-reinforced unsaturated polyester (named GF/VE and GF/UP, respectively) were chosen to suffer rigorous hygrothermal aging. Their mechanical performance was monitored during the aging process to evaluate their durability. The cause of deterioration of the composite was comprehensively analyzed. Based on the analysis results of attenuated total-reflectance-Fourier-transform infrared spectroscopy (ATR-FTIR), thermogravimetric analysis (TGA) and dynamic mechanical analysis (DMA), the change mechanism of chain structure of the resin molecule was proposed. SEM (scanning electron microscopy), X-ray photoelectron spectroscopy (XPS) and X-ray diffraction (XRD) were used to analyze the microstructure and degradation mechanism of the fiber and the interface between fiber and matrix. The degradation mechanism of the composite system, including the resin, the fiber and the interface, was obtained, and it was found that the deterioration of the matrix resin caused by the hygrothermal environment was the main factor leading to the decline in composites performance.

## 1. Introduction

Epoxy vinyl ester (VE) and unsaturated polyester (UP) thermosetting resins reinforced with glass fiber, are two materials frequently used in a growing number of applications, because of their light weight, high strength, electromagnetic transparency, low maintenance and low price [1,2,3]. There are already some bridges, buildings, equipment and water tanks that are comprised of these two structural elements [4,5]. Both composite applications will continue to grow globally at robust rates in following years [6,7]. Many of these kind of materials are exposed to a relatively corrosive or aggressive environment [8]. Equipment and structure are, especially in a moist tropic environment, corroded by the cooperation of heat and moisture, which accelerates the deterioration of the composite [9].

Research of their long-term performance provides empirical evidence of this deterioration and degradation. Jefferson et al. [10,11,12] study the influence of hygrothermal aging on carbon nanofiber/fiber-enhanced polyester/epoxy material systems. Ghabezi et al. [13,14,15] investigate the adding effect of nanoparticles grade on filling time in the vacuum-assistant-resin-transfer molding process by producing composite samples made of glass fibers and epoxy resin. Dionysis et al. [16,17,18] assessed the failure of polyester/glass fiber-reinforced composites when exposed to the combined action of temperature, humidity and a UV radiation environment. Yang et al. [8,19,20] proposed a prediction model to describe the impact of degradation trends and temperature effects on the ultimate bearing capacity of GFRP composites in hygrothermal environments. Dhakal et al. [21,22,23,24,25] demonstrated the effect of water absorption on the mechanical properties of fiber-reinforced unsaturated polyester/other plastics composites. Visco et al. [26,27] found that polyester resin was more degraded than the vinyl counterpart, due to a different network organization of the two materials in seawater immersion.

Therefore, the study of the failure mode in the hygrothermal environment is of the utmost importance for composite failure research. Present studies of these composites either merely focus on the degradation analysis of the matrix, or on the whole changes of mechanical performance. Moisture in the presence of heat affects the matrix, the reinforced fiber and the interface, and causes the decline of composite performance [28,29,30]. Complex research of the composite system, including of changes in the matrix, reinforced glass fibers and the interface, is essential and urgent. Furthermore, comprehensive and validated data on the durability of these two composites (GF/VE and GF/UP) are still scarce, and there is a lack of comprehensive research and understanding of the aging and degradation mechanisms that refer to the hygrothermal service conditions they are likely to be subjected to.

It is essential to investigate their long-term performance, deterioration mechanism, and the key factors affecting their mechanical behavior in an accelerating hygrothermal environment, as this will provide empirical evidence of their selection and maintenance. In order to assess the durability of the composites in an aggressive tropic environment, find their weakness of position, investigate the key factors affecting their mechanical behavior, and understand the regular change of the composite system, a new investigation method of the fiber was employed. This article monitored their mechanical performance, characterized the micromorphology of matrix, fiber and interface, and analyzed the change mechanism of the resin molecule’s chain structure during the hygrothermal aging process. The comprehensive analysis demonstrated that deterioration of the matrix resin mainly led to the decline in composite performance.

## 2. Materials and Methods

### 2.1. Materials

Two composites made of EKB45 E-glass fiber orthogonal plain weave fabric (0.450 g/m^2^, Changzhou Tianma Group Co., Ltd., Changzhou, China) and 905-2 epoxy vinyl ester (1.15 kg/L, 380 cps, tensile strength 84 MPa, flexural strength 122 MPa, Swancor Advanced Materials Co., Ltd., Shanghai, China) named as GF/VE, and EKB45 E-glass fiber orthogonal plain weave fabric (0.450 g/m^2^, Changzhou Tianma Group Co., Ltd., Changzhou, China) and 197S unsaturated polyester resin (1.09 kg/L, 350 cps, tensile strength 57 MPa, flexural strength 115 MPa, Swancor Advanced Materials Co., Ltd., Shanghai, China) named as GF/UP, were fabricated.

The following outlines the brief process of making the composite samples. Two kinds of thickness glass fiber-reinforced resin composites were prepared by using the vacuum-assisted molding process at room temperature. The thicknesses are are 2 mm and 6 mm, respectively. The fiber content in GF/905-2 is 62 wt%, and in GF/197S is 60 wt%. Subsequently, the composite panels were cut into specimens for tension, bending, interlaminar shear, and compression testing, by using the water cutting method in accordance with the corresponding test standards.

### 2.2. Methods of Hygrothermal Aging and Immersion

Hygrothermal aging was performed in a Yinhe SH050 (Chongqing Yinhe Test Instrument Co., Ltd., Chongqing, China) hygrothermal aging chamber with a condition of 70 °C and 95% RH, according to the standard of GB 2574-1989 [31]. Four types of composite samples which will be subjected to tensile, flexural, compressive, and shear mechanical testing were performed by hygrothermal aging. The composite plates were cut into specimens by water cutter and used directly without sealed edges. Specimens were tested at 10 d, 21 d, 46 d, 79 d, 108 d, 133 d, 193 d, 258 d, 333 d, 400 d for all intervals.

Five pieces of the samples were tied together by a string, and immersed and suspended in pure water at room temperature. The dimensions of the specimens used in mass change rate examination were the same as those of the compressive specimens. Sample mass was weighed on a SHIMADZU UW1020H balance (Kyoto, Japan).

### 2.3. Mechanical Properties Testing

Tensile strength and flexural strength were tested on a Reger RGT-10A universal electronic testing machine (Upshur County, WV, USA). Compressive and shear strength were tested on a Shanghai WE-10B universal hydraulic testing machine (Shanghai, China). The dimension of the specimens and testing set-up was used by previous studies [7].

### 2.4. Testing and Characterization

ATR-FTIR analysis was conducted with a Perkin Elmer Spectrum 400 ATR-FTIR instrument (Waltham, MA, USA) with 4 cm^−1^ resolution.

XPS was collected using an Ulvac-Phi PHI 5000 VeralPro XPS instrument (Chigasaki, Japan) with an Al: Kα excitation source, a resolution ≤ 10 μm, and a vacuum degree ≤ 6.7 × 10^−10^ mbar.

Microtopography was performed with an FEI QUANTA 200 SEM (Hillsboro, OR, USA) with 3.5 nm resolution, with accelerating voltage ranging from 200 V–30 kV.

An XRD spectrum was performed on a D8 Advance XRD machine (Billerica, MA, USA) with a graphite monochromator. Kα ray was produced by a solid copper target with a 5°/min scan speed. Voltage and current were set at 40 kV and 40 mA, respectively.

In order to find the exact impact of the environment on the viscoelastic behavior of the materials, thermal scans were performed on small bars of ca. 60 × 10 × 2.5 mm^3^ for the composite with a DMA242 Netzsch instrument (Selb, Germany). The test mode was three points bending under controlled strain. The temperature was set between 30 and 200 °C, with a constant frequency of 0.5 Hz during temperature scans. Preloading was set at 2.0 N, and the strain amplitude was 30 µm.

In order to find the exact impact of the environment on the thermal degradation behavior of the materials, the mass loss of samples was recorded during their thermal aging. The TGA cell was performed on a STA 449C Netzsch instrument (Selb, Germany) in an inert (100% N_2_) atmosphere.

## 3. Results

### 3.1. Mechanical Properties

Figure 1, Figure 2, Figure 3 and Figure 4 show the evolution of the mechanical properties of the GF/VE and the GF/UP composites in a hygrothermal aging environment. The hygrothermal environment significantly affects the mechanical performance of the samples, including tensile, compressive, flexural, and shear strength. It has a more sustained and serious effect on the GF/UP composite than on the GF/VE. For GF/UP, the mechanical performances of four types severely dropped by 40–60% after aging for 400 days. However, for GF/VE, tensile and flexural strength dropped by about 30%, and compressive and shear strength only dropped slightly after 400 days of aging. All these changes indicate that the GF/UP composite is more sensitive to the hygrothermal environment than GF/UP. The reason for the differences in the decline of mechanical performance may be ascribed to the difference in the matrix resin structure. The two ester groups are in close proximity, and being ortho to each other minimizes the crowdedness of the UP esters. The aromatic esters cannot possibly rotate to any transforms and are therefore in a state with a high strain energy. In order to release this energy, the UP esters would prefer to ‘kick-off’ the esters. Meanwhile, the incomplete double bond, when the composite was prepared at room temperature, is easily oxidized at high temperature. This is why UP, compared to EV, are more prone to attack by hydrolysis, and it is also why UP resins will fail in the hygrothemal type of aging, especially at an elevated temperature.

### 3.2. Appearance and Failure Surface Morphology Observation

Figure 5 shows the appearance and destroyed macrostructure of the GF/VE and GF/UP composites in hygrothermal aging conditions. The hygrothermal environment significantly effects appearance and macrostructure failure in both the GF/VE and the GF/UP composites. The surface of the two aged composites turned more yellow and darker in the entire procedure of the artificial weathering test. For GF/VE, after 46 days of aging, the surface color changed greatly and became yellow. After aging for 333 days, the surface color changed to brown. The failure mode turns from simple fiber break to fiber break combined with fiber pull out. For the GF/UP, after aging for 46 days, the surface color changed greatly and became more yellow and the tensile failure mode was a combination of fiber break and resin cracking. After 333 days of aging, the surface color changed to brown and the tensile failure mode became a single fiber break.

### 3.3. Thermal Properties

#### 3.3.1. DMA Analysis

Figure 6 and Figure 7 show the DMA curves of the GF/VE and the GF/UP composites during hygrothermal aging. For both composites, the storage modulus decreased and Tg increased with aging time. Storage modulus of the GF/VE increased slightly, from 8300 MPa to 9700 MPa, after aging for 258 days and the Tg increased from 91 °C to 96 °C. After aging for 333 days, the storage modulus decreased to 5300 MPa and Tg increased to 97 °C. The storage modulus of GF/UP decreased remarkably from 11,800 MPa to 8000 MPa after aging for 258 days, and the Tg increased from 106 °C to 111 °C. After aging for 333 days, the storage modulus decreased to 7000 MPa and Tg increased to 113 °C. The decrease in storage modulus can be attributed to the plastic effect of water entering into the composite, which was proven by a water absorption test. The increase in Tg can be attributed to the post cure effect of unsaturated bonds, which was proven by the FTIR absorption spectra shown in Figures 10–13.

#### 3.3.2. TGA Analysis

Figure 8 shows the TGA curves of GF/VE and GF/UP composites in the hygrothermal condition after aging for 333 days. For both composites, there is a small mass loss of about 3% from 120–300 °C, which may be attributed to the volatilization of the small molecules in the resin. The rapid weight loss above 300 °C, meanwhile, can be attributed to the degradation of the basic resin. It is worth noting that degradation temperature slightly increased for the hygrothermal-aged sample of GF/UP composite, which further confirmed the post-cured process of this basic resin in hygrothermal conditions.

### 3.4. Mass Change Rate

The two composites were immersed in pure water at room temperature to investigate their mass change, as shown in Figure 9. The two composites absorb water quickly at an early stage, which is the mass increased linearly–mass increase stage (Stage I). After 50 days of immersion, the mass basically kept the same mass stability stage (Stage II). Up until 120 days, the mass began to decrease again and then, after being continuously immersed for 30 days, the mass decreased to a low point, mass decrease stage (Stage III). From 170 days, the mass increased greatly again, in a second mass increase stage (stage IV). From Stage I to Stage IV, the physical and chemical reaction process in the composites occurs as follows: in Stage I, water enters into the micro voids and defects purely, which leads to mass increase; in Stage II, part of small molecules are gradually dissociated from the matrix into the absorbed water; in Stage III, the composite material is saturated with water. However, the small molecule is continually diffused out of the composite, which cause the decrease of the whole mass; in Stage IV, more water enter into the composite micro voids due to the diffusion of the small molecule. The absorption of water generates a plastic effect that acts on the composite, which is proved by the decrease of the storage modulus in DMA.

## 4. Discussion

### 4.1. Changes of the Matrix Resin

#### ATR-FTIR Analysis

Figure 10 shows the ATR–FTIR spectra of the GF/VE composite surface during hygrothermal aging. The spectra indicated that the band intensity, at 898 cm^−1^ and 875 cm^−1^, was due to the saturating C-H deformation increasing after 108 days of aging, clearly demonstrating further linkage of the residual double bond and post curing of a small part of the matrix resin in this period.

Based on the above analysis results for ATR–FTIR, DMA, and TGA, it is deduced that a post cure reaction of the linkage of residual double bonds occurred in the VE resin in hygrothermal conditions, as shown in Figure 11. An evident increase in Tg and thermal degradation temperature also indicates that the post cure process has taken place. Although there is a weak bond in the polymer (which is asterisk marked) that can be attacked by oxygen or water at high temperature, there is no obvious oxidation and degradation phenomenon during the aging process in the resin. But in high temperature conditions, water molecules entered into the composites and plasticization of the basic resin occurred.

Figure 12 shows ATR–FTIR spectra of the GF/UP composite surface during hygrothermal aging, confirming a noticeable change in the spectrum. After 46 days of aging, absorbance at 1644 cm^−1^ and 1607 cm^−1^ that correspond to th C=C double bonds region disappears, indicating further linkage of the residual double bond in this period. During the whole aging process, absorbance at 1151 cm^−1^, corresponding to deformation of C-OH, continuously decreased, and absorbance at 1178 cm^−1^ and 742 cm^−1^, corresponding to C-O ester, increased, indicating further etherification of end hydroxyl.

Based on ATR-FTIR analysis and the preceding analysis of results for DMA, TGA, shear and compressive strength of the GF/UP, it is deduced that a post cure reaction occurred in the UP resin, including continuous linkage of residual double bonds and esterification of end hydroxyl in hygrothermal conditions, as shown in Figure 13. There is a weak bond in the polymer (marked by an asterisk) that can be attacked by oxygen or water at high temperature. There is no obvious oxidation and degradation phenomenon during the aging process in the resin. But in the high temperature condition, water molecules entered into the composites, and plasticizing of the basic resin occurred, which may have caused the decrease of the storage modulus in DMA.

The yellowing color of the resin reveals that incompletely reacted phenol and amine compounds are oxidized to quinone by the combined action of oxygen and high temperature. In addition, the yellow transformation of the composite appearance can be attributed to the oxidation of p-dihydroxybenzene, both as an inhibitor and anti-oxidant, to benzoquinone in an oxygen atmosphere.

### 4.2. Changes of the Glass Fiber

In order to investigate the effect of aging on glass, high temperature air combustion technology was used to remove the matrix resin from the fiber surface of the composite samples after aging for 400 days. The appearance, elements, and crystal form of the glass fibers were subsequently observed and analyzed.

#### 4.2.1. Appearance Observation

As shown in Figure 14, which provides a macrophotograph of fiber, the color of UP-based fiber turns to dark grey from a nearly transparent color, while VE-based fiber retains the original color after aging for 400 d. The changes in fiber color indicate that fiber in the UP resin-based composite have undergone some transformation. XPS and XRD tests were used to further investigate the cause of this change.

#### 4.2.2. XPS Analysis

Figure 15 presents the fiber in the UP resin-based composite, showing that the binding energy of silicon increased from 101.92 eV to 102.39 eV, indicating an increase of the average valence of silicon, which can be attributed to the water molecule invading the fiber network. At high temperature, H_2_O entered into the Si-O-Si structure, destroyed part of the network structure and produced silicate, making the average valence of silicon increase. In the fiber in the VE resin-based composite, the average valence of silicon basically remains the same. For both UP and VE based fibers, there is no obvious change in the binding energy of calcium.

#### 4.2.3. XRD Analysis

XRD spectra of the fiber, both before and after aging, are shown in Figure 16. A peak characterizing crystalline phase appears neither in the spectrum of fiber enwrapped by VE (Figure 16a), nor in the spectrum of fiber enwrapped by UP (Figure 16b), indicating that the hygrothermal aging condition did not obviously change the fiber’sphase state.

#### 4.2.4. Microstructure Observation

SEM scans provided evidence of the environmental impact on the fiber and interface in the composite specimens. Figure 17 depicts the failure interface in the tensile fracture section of the glass fiber in the GF/VE composite, both in the original and weathering aged specimens. Partial microcracks and microvoids can be seen on the surface of tensile failure specimens after 400 days of aging. These microdefects can lead to brittleness of the fiber and possibly affect the tensile strength of the composite. Some fiber ruptured along the radial direction when the delamination interface was examined, as indicated by the arrow in the figure. When combined with the analysis results of SEM and the mechanical properties, this leads us to conclude that the strength of glass fiber decreased because of the effects of the hygrothermal environment.

Figure 18 depicts the virgin and weathering aged surface of the glass fiber tensile failure specimens of the GF/UP composite. Several fibers were broken off after 400 days of hygrothermal aging, as shown in the bottom right micrograph in Figure 18, which indicates that hygrothermal environment promotes the deterioration of the glass fiber. On the basis of the XPS results and the mechanical properties, it can be concluded that the strength of glass fiber decreased in hygrothermal condition.

##### Summary of Deterioration Reason for Glass Fiber

The main ingredients of E-glass fiber are as follows: 53.55 SiO_2_, 13.16 Al_2_O_3_, 16.23 CaO, 0.5 MgO, 10 B_2_O_3_, 0.12 Fe_2_O_3_, 0.5 TiO_2_. The SiO_2_ forms the net framework structure, and other alkali metals disperse in this framework. A series of chemical reactions may occur in this kind of glass fiber in acidic conditions. As a result, the strength of the fiber decreases when subject to the combined influence of aging conditions and tensile stress. Corrosion of the glass fibers will impact the performance of glass fiber- reinforced composites. In this experiment, the decline of the fiber performance can be concluded on the basis that there are some micro defects in the basic resin and in the interface between the resin and glass fiber. Water molecules therefore diffuse into these defects, improve corrosion of the glass fiber, and lead to a decrease in the composite system’s performance.

### 4.3. Changes of the Interface

The traversed cutting fracture surfaces and interface between the fiber and resin of GF/VE tensile failure specimens, in both pristine and aged conditions, are shown in Figure 19. It can be seen that, after aging for 108 days, when the composite was destroyed by the tensile method, there was almost no resin enwrapped in the fiber. The interfacial debonding phenomena was therefore first exhibited at this aging stage, and there was no evident change in the glass fiber surface. However, after aging for 193 days, when the composite was split at a longitude direction, some cracks appeared in the matrix resin. After aging for 333 days, there was no evident change in the matrix resin. The composite was then broken off by a shear in the vertical direction. An analysis of the fracture surface shows the exhibited debonding phenomena was trivial after aging for 133 days as showed in Figure 19b (black arrows indicated). The evolution of debonding and fiber cracking phenomena was more obvious after aging for 333 days as showed in Figure 19b (black arrows indicated). However, glass fiber cracked along the long direction on the fracture surface after aging for 333 days, revealing that the tensile stress that destroys the composite also leads to a crack emerging in the matrix resin after aging for 193 days. But the tensile stress that destroys the composite will not lead to the matrix resin cracking after 333 days of aging.

The SEM microphotographs of fracture surfaces and the interface between the fiber and resin of the GF/UP tensile specimens, in both pristine and aged conditions, are shown in Figure 20. As seen, most of the basic resins still adhered to the glass fiber after aging for 46 days, when the composite was broken by the splitting method. There was no evident change in the interface. However, after aging for 193 days, when the composite was split by longitude portrait, substantial resin deboned from the fiber. In referring to the fracture surface, we can see that cracks formed on the interface and that the interfacial debonding phenomena formed during the aging process. However, the debonding phenomena became more visible, and some cracks appeared in the matrix resin after aging for 333 days as showed in Figure 20b (black arrows indicated). We therefore conclude that hygrothermal conditioning promoted debonding of the interface and destroyed the interface, which also reveals that the tensile stress that destroys the composite also leads to the matrix resin cracking, after aging for 333 days.

In referring to changes in the interface photograph of the two composites, we see that the hygrothermal environment promoted bonding of the interface and destroyed the interface at an early stage. As aging occurred over time, the fiber became brittle, and could be more easily destroyed by stress than the interface. With regard to the degeneration speed, GF/UP is seen to be faster than GF/EV.

## 5. Conclusions

In this study, the mechanical properties of two glass fiber-reinforced composites were analyzed by referring to test data. The cause of regular changes in the mechanical performance and probability degradation mechanism of the composite system was proposed. The following conclusions were drawn:The hygrothermal environment causes performance decline, both in GF/VE and GF/UP composites. The hygrothermal environment has a stronger effect on GF/UP than on GF/VE.High temperature, when combined with water, has significant effects on the matrix resin and reinforcing fiber of the GF/UP composites system, despite the fiber being wrapped and bonded by the matrix resin. However, this environment mainly affects the basic resin of the GF/VE composites system, and basically does not affect the fiber. The degeneration mechanism of E-glass fiber should be investigated further.Analysis of the results for SEM, ATR-FTIR, DMA, TGA and mechanical properties, indicate that the degradation of the matrix resin and glass fiber are mutually promoted. The cooperating effects of heat and humidity cause the degradation of the composite system, both in the resin and the interface.On the one hand, the carbolic acid obtained from the hydrolysis of the resin erodes the glass fiber and accelerates its degradation. On the other hand, the product from the glass fiber corrosion promotes hydrolysis and accelerates the degradation of the resin.The interaction between the basic resin and reinforced fiber should be further researched. Aging mechanism and failure mode of the fiber, along with the interface between the fiber and the matrix in the aforementioned two systems, should be the primary preoccupation of further study and research.

## Figures and Tables

**Figure 1 polymers-16-00632-f001:**
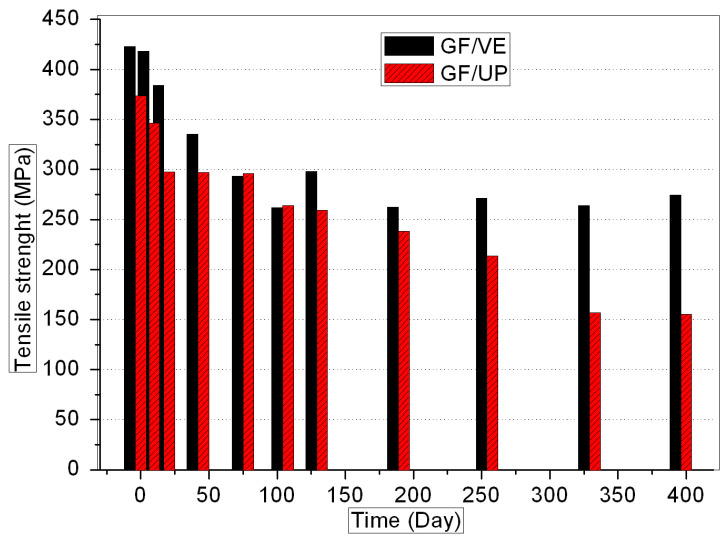
Tensile strength versus hygrothermal aging time for the GF/VE and the GF/UP composites.

**Figure 2 polymers-16-00632-f002:**
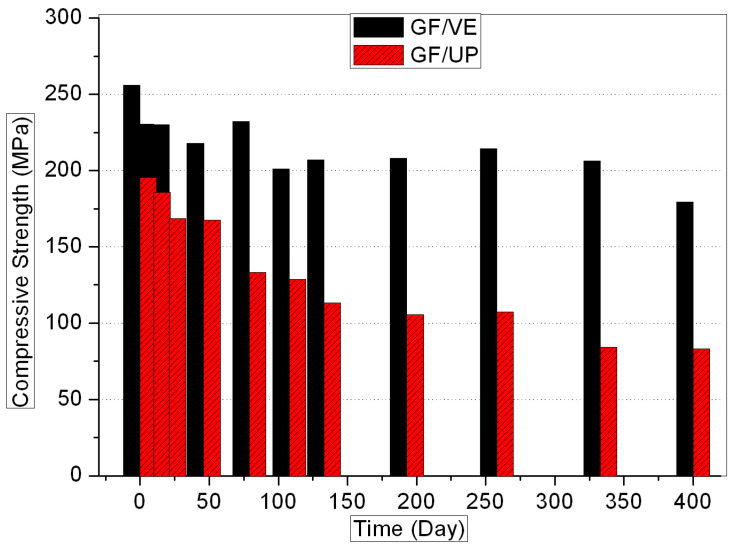
Compressive strength versus hygrothermal aging time for the GF/VE and the GF/UP composites.

**Figure 3 polymers-16-00632-f003:**
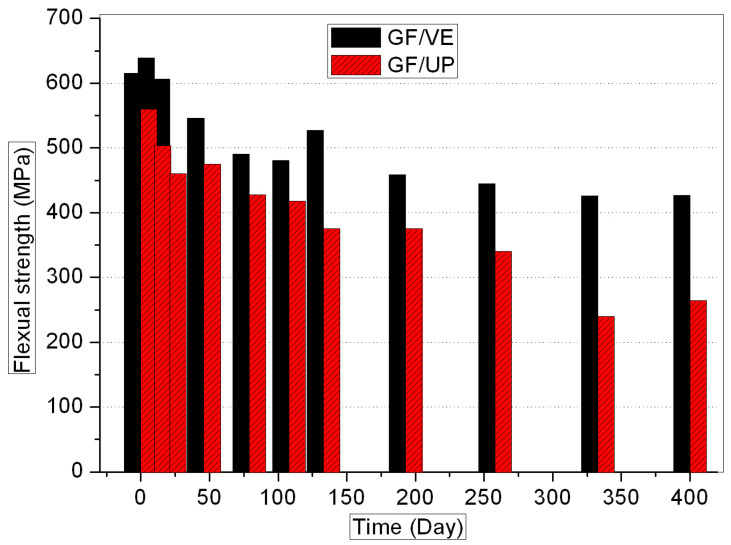
Flexural strength versus hygrothermal aging time for the GF/VE and the GF/UP composites.

**Figure 4 polymers-16-00632-f004:**
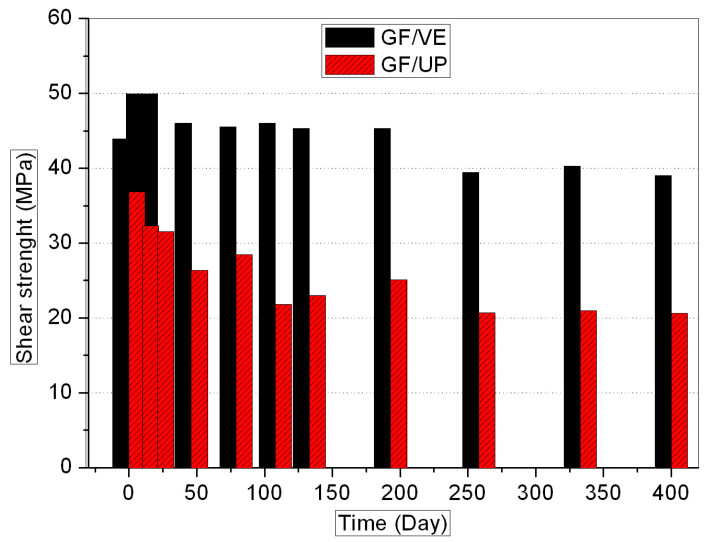
Shear strength versus hygrothermal aging time for the GF/VE and the GF/UP composites.

**Figure 5 polymers-16-00632-f005:**
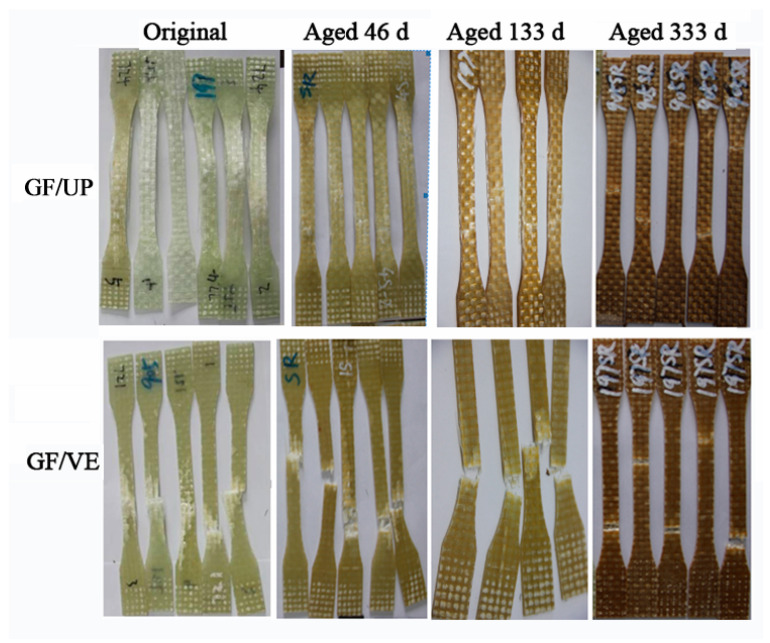
Appearance and macrostructure of the tensile failure GF/VE and GF/UP composites during hygrothermal aging.

**Figure 6 polymers-16-00632-f006:**
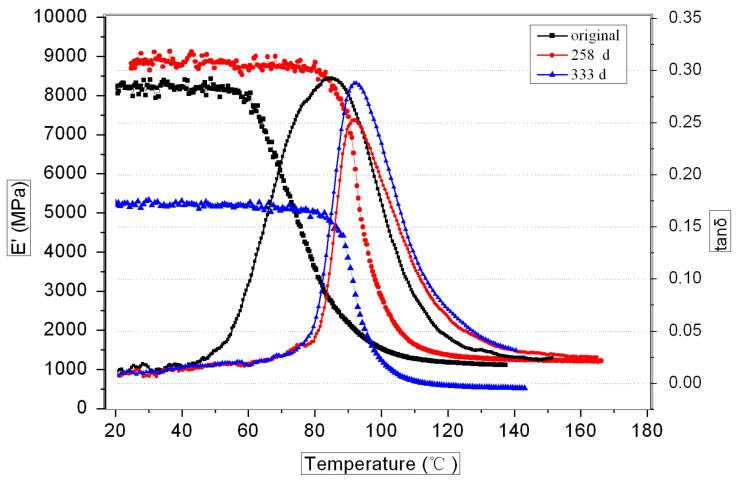
DMA curves of the GF/VE composite during hygrothermal aging.

**Figure 7 polymers-16-00632-f007:**
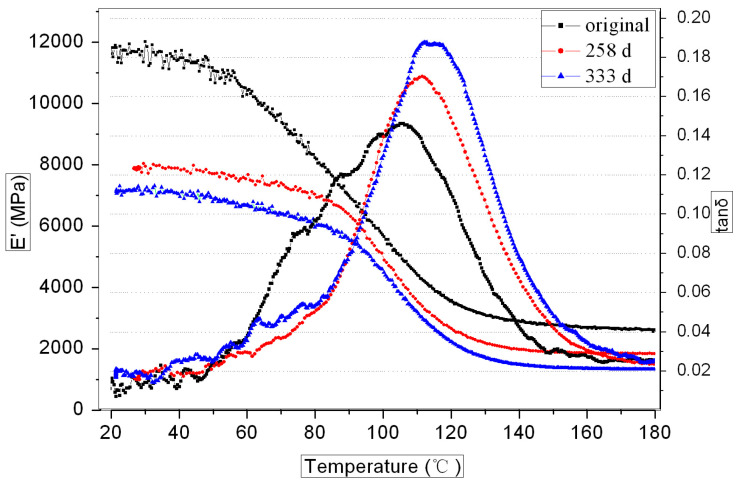
DMA curves of the GF/UP composite during hygrothermal aging.

**Figure 8 polymers-16-00632-f008:**
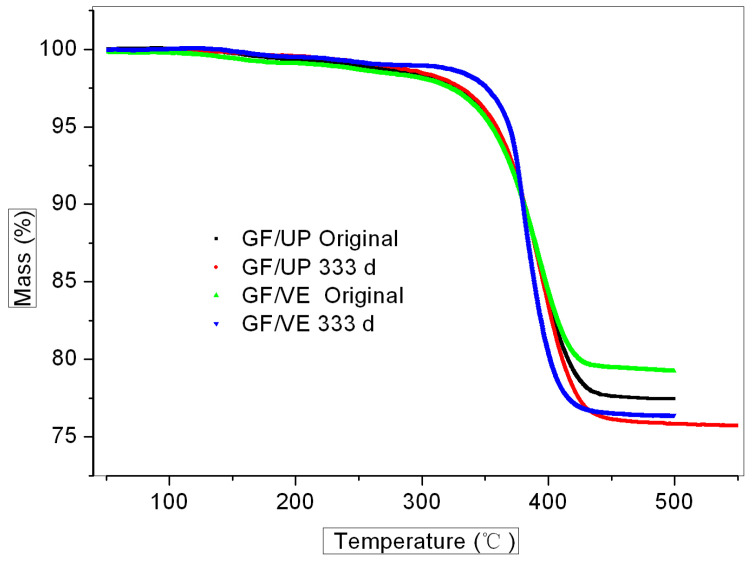
TGA curves of the GF/VE and the GF/UP composites, after 333 days of aging in the hygrothermal aging condition.

**Figure 9 polymers-16-00632-f009:**
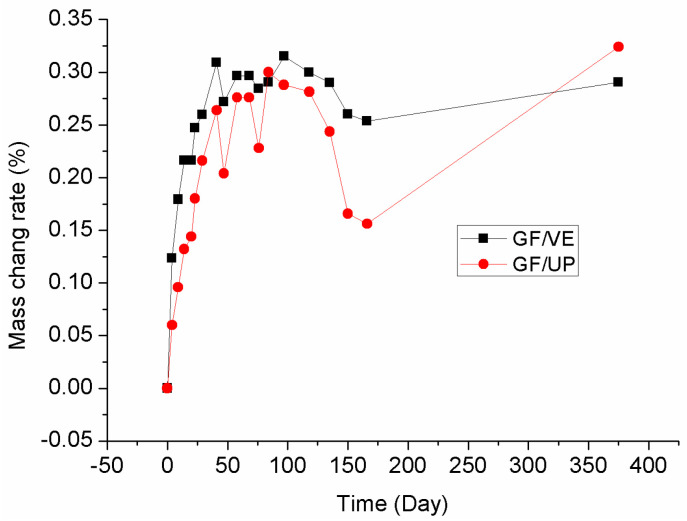
Mass change rate curves of the GF/VE and the GF/UP composites during immersion aging at room temperature.

**Figure 10 polymers-16-00632-f010:**
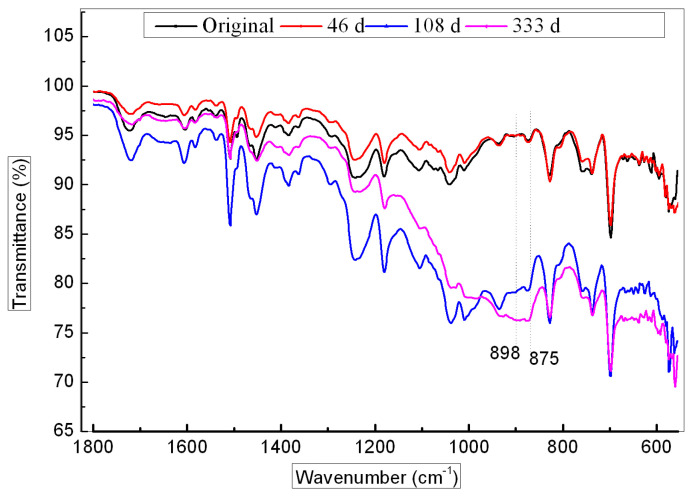
ATR–FTIR spectra of the GF/VE composite surface during hygrothermal aging.

**Figure 11 polymers-16-00632-f011:**
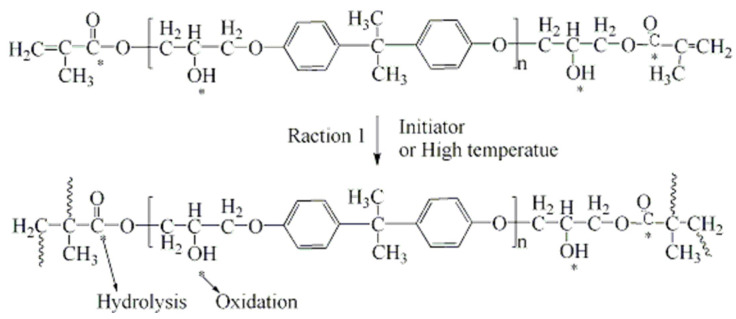
Cross-link reaction of the VE resin prepolymer to its weak bond (indicated by an asterisk).

**Figure 12 polymers-16-00632-f012:**
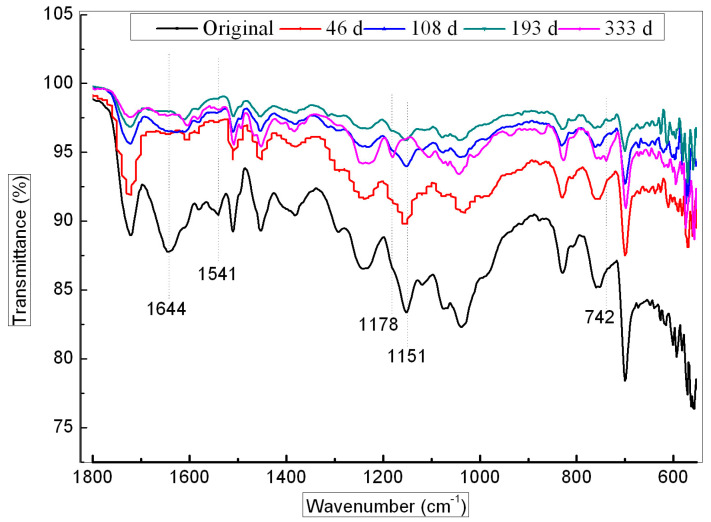
ATR–FTIR spectra of the GF/UP composite surface during hygrothermal aging.

**Figure 13 polymers-16-00632-f013:**
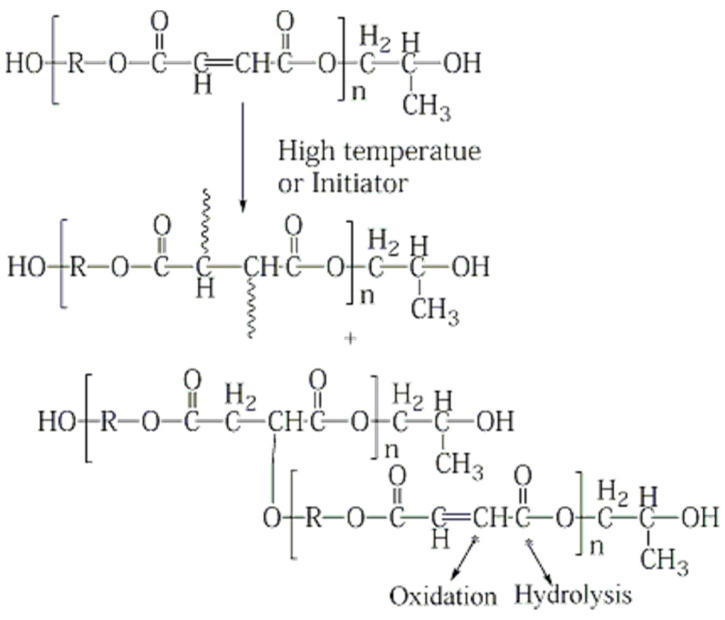
Cross-link reaction of the UP resin prepolymer to its weak bond (indicated by an asterisk).

**Figure 14 polymers-16-00632-f014:**
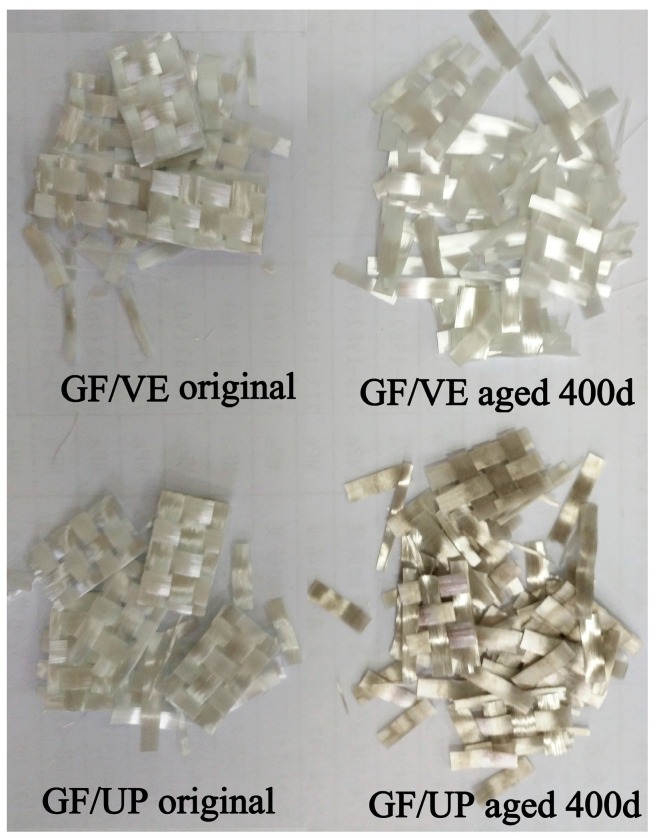
Appearance of the glass fiber after the resin is burned out.

**Figure 15 polymers-16-00632-f015:**
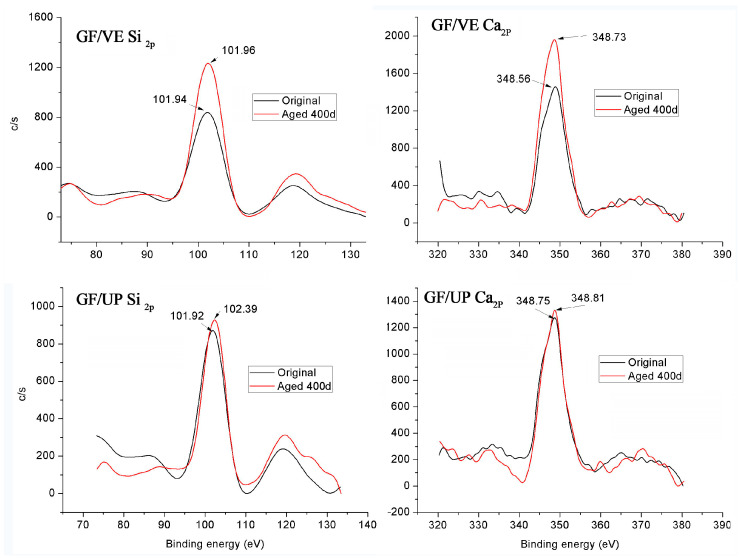
XPS spectra of silicon and calcium in the glass fiber.

**Figure 16 polymers-16-00632-f016:**
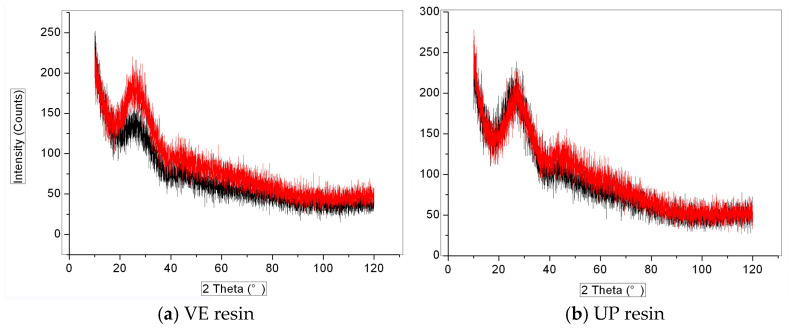
XRD spectra of the glass fiber (**a**) VE resin, (**b**) UP resin (curves of black color: original samples; curves of red color: aged samples).

**Figure 17 polymers-16-00632-f017:**
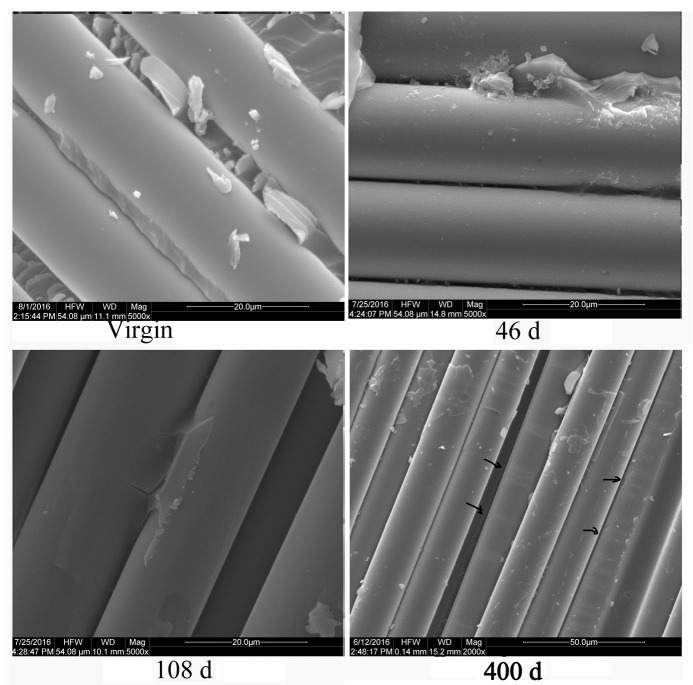
SEM microphotographs of the glass fiber, showing tensile failure in the GF/VE composite during hygrothermal aging.

**Figure 18 polymers-16-00632-f018:**
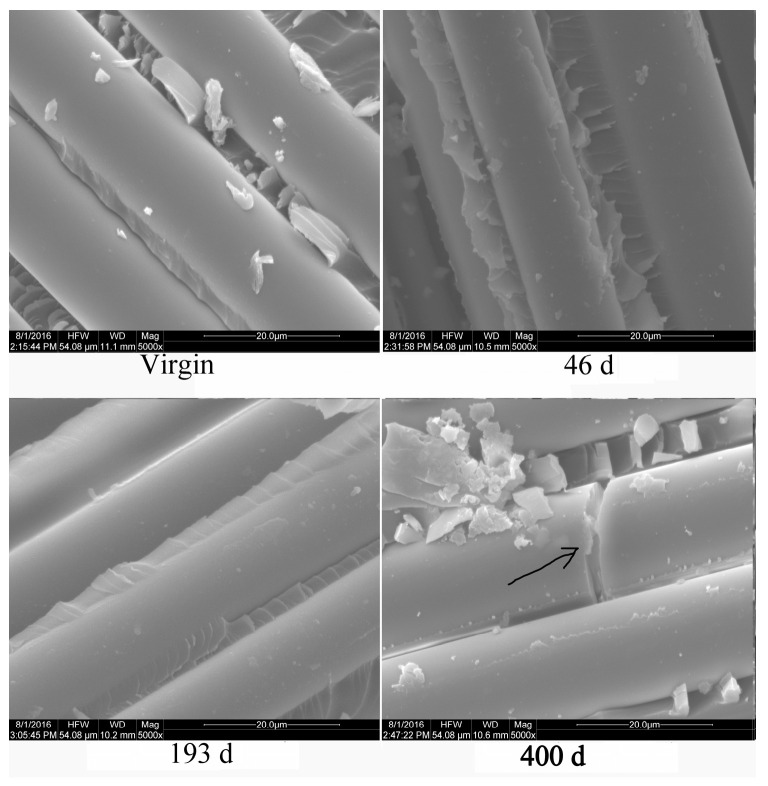
SEM microphotographs of the glass fiber, showing tensile failure in the GF/UP composite during hygrothermal aging.

**Figure 19 polymers-16-00632-f019:**
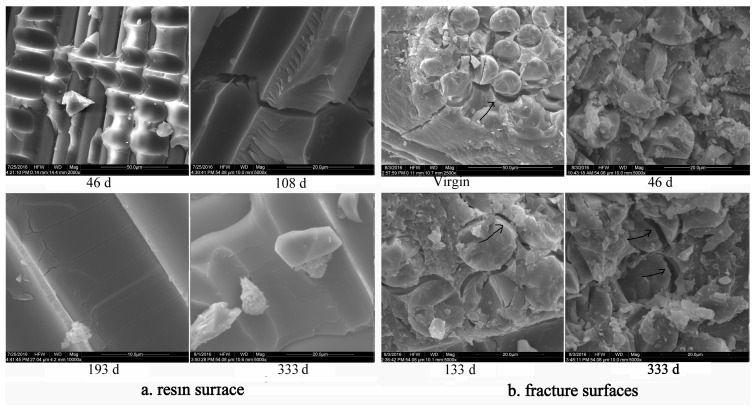
SEM microphotographs of the GF/VE composite during hygrothermal aging (**a**) resin surface after splitting; (**b**) fracture surfaces after cutting off.

**Figure 20 polymers-16-00632-f020:**
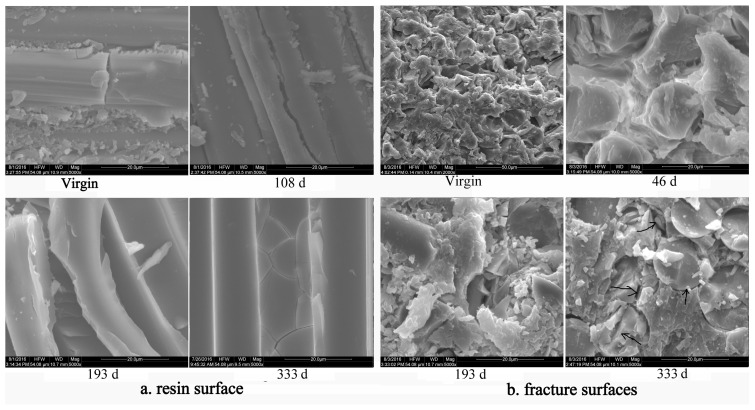
SEM microphotographs of the GF/UP composite during hygrothermal aging (**a**) resin surface after splitting; (**b**) fracture surfaces after cutting off.

## Data Availability

Data are contained within the article.

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
