# Peer review of "Hygrothermal Effect on GF/VE and GF/UP Composites: Durability Performance and Laboratory Assessment"

_polymers, 2024, doi:10.3390/polym16050632_

Round 1
Reviewer 1 Report
Comments and Suggestions for Authors
1. Can the author provide more details about the fabrication process of the composites, such as the specific vacuum molding technique used and any parameters or conditions that were controlled during the fabrication process? The fractions of fibers in relation to matrixes used in the elaboration of composites?
2. It would be beneficial to provide in the section of materials and methods the information on the mechanical properties of the raw materials (E-glass fiber, unsaturated polyester resin, and epoxy vinyl ester) used in the fabrication of the composites?
3. Can the author explain with more detail in the mechanical proprieties section why the GF/UP composite is more severely affected by the hydrothermal environment than the GF/VE composite? Are there any specific chemical or structural factors that contribute to this difference?
4. Could the author add a paragraph in the discussion section to provide comparisons with existing literature or similar studies that have investigated the effects of hydrothermal aging on similar composite materials?
5. Instrument generated image information for SEM micrograph (Figure 17, Figure 18, and Figure 19) is insufficient for publication. Please crop this from the micrographs provided and place a clearly visible scale bar on the images. Any imaging information that is required for the reader should be provided in the corresponding Figure caption.
Reviewer 2 Report
Comments and Suggestions for Authors
· Do not use abbreviated words in the abstract.
· In the abstract, you stated that “deterioration of the fiber caused by the hygrothermal environment was a main factor leading to the decline in composites performance.” What do you mean by “deterioration of the fiber”? Do you think that the fibres were degraded?
· The novelty of the work shall be highlighted in the last paragraph of the introduction section.
· In the last paragraph of the introduction section, what is done, how, and what was found should be presented.
· What is the merit, and motivation for studying E-glass fiber-reinforced composite materials (GF/VE and GF/UP)?
· The introduction is too short and needs to be strengthened by adding more recently published papers in this area. There are many other publications discussing the hygrothermal process in epoxy-based composite materials (mechanical properties, multi-scale modeling, life prediction, etc.) at elevated temperatures. I recommend you review some of the recently published papers and summarize them in the introduction section.
· Page 1, “The present studies on these composites either merely focus on 39 the degradation analysis of the matrix, or on the whole changes of mechanical performance”. You need to discuss those works not just generally refer to them or you can summarize them including those above-mentioned in a table in terms of materials, temperatures, medium, etc.
· The materials section is very poor. More details regarding the used materials must be added such as mechanical properties, sources, etc.
· Some samples are aged in a chamber with conditions of 70℃ and 95% RH and some are immersed in water. Why??? How can you compare the results together while they were aged in different environments?
· The elevated temperature in the chamber was 70℃. Did you check the glass transition temperature of the polymers?
· The samples were tested in a chamber with 70℃ and 95% RH, while the others were tested in water at room temperature. How do you want to make a connection between these different conditions? You cannot compare the results gathered from two different media.
· The section “2.2. Methods of hygrothermal aging and immersion” must include more details.
· What standard test method did you employ to perform the aging process?
· Were the edges of the samples sealed?
· What about the dimensions of the test samples?
· What about the water uptake samples?
· The sample's fibre volume fractions?
· How did you conduct the tests in section “2.3. Mechanical Properties Testing”? Standards, test speed, conditions, sample dimensions, etc.
· There are no details regarding the vacuum bagging process of the samples. Refer to these works and add more necessary details including a Figure. 1- Investigation of Mechanical Behavior of Alfa and Gamma Nano- Alumina/Epoxy Composite Made By VARTM, 2- Study on Mechanical Properties of Carbon Nanotube Reinforced Composites and 3- Experimental investigation of nano-alumina effect on the filling time in VARTM process
· How did you perform the shear test?
· Figure 5, shows composite samples with dog-bone shapes. Are you sure that you can test the composite samples reinforced with fibres in a dog-bone shape? What test standard do you follow? According to ASTM3039, the test samples shall be rectangular.
· Did you use tabs for tensile testing?
· More details regarding the materials and testing must be added.
· The discussion section is poor.
· In Figure 9, why there is a drop in the mass of samples after almost 150 days?
· Did the samples reach their saturation point?
· Use bullets to highlight the main achievements of the work in the conclusion section.
Round 2
Reviewer 2 Report
Comments and Suggestions for Authors
The paper is accepted in its current form.